# BOOSTAPR: Boosting Automated Program Repair via Execution-Grounded Reinforcement Learning with Dual Reward Models

Yuanhao Li [1]  Hongbo Wang [1]  Xiaotang Shang [1]  Xunzhu Tang [2]  Yiming Cao [1]  Xuhong Chen [1]

## Abstract

Reinforcement learning for program repair is hindered by sparse execution feedback and coarse sequence-level rewards that obscure which edits actually fix bugs. We present BOOSTAPR, a three-stage framework addressing these challenges: (1) supervised fine-tuning on execution-verified demonstrations with reasoning traces, (2) training dual reward models—a sequence-level assessor and a line-level credit allocator—from execution outcomes, and (3) PPO optimization where the line-level model redistributes rewards to critical edit regions. This line-level credit assignment operates at an intermediate granularity naturally suited to code changes. Trained on SWE-Gym and evaluated on four benchmarks, BOOST-APR achieves 40.7% on SWE-bench Verified (+22.9pp over base model), 24.8% on Defects4J (Python→Java transfer), 84.5% on HumanEval-Java, and 95.0% on QuixBugs, achieving competitive results among open-source models with strong cross-language generalization.

## 1. Introduction

Automated program repair (APR) represents one of the most consequential applications of artificial intelligence to software engineering, with the potential to dramatically reduce the substantial human effort devoted to debugging and maintenance (Le Goues et al., 2019; Monperrus, 2018). The emergence of large language models (LLMs) has catalyzed remarkable progress in this domain, enabling systems that can generate patches conditioned on rich contextual signals including bug descriptions, failing test outputs, and repository-wide code structure (Xia et al., 2023; Jiang et al., 2023a; Yang et al., 2024; Xia et al., 2024; He et al., 2026).

Despite this progress, several fundamental challenges continue to limit the effectiveness of LLM-based APR systems. First, **execution feedback is inherently sparse**: a patch either resolves all tests or it does not, providing a binary signal that offers limited guidance for learning. Unlike domains such as text generation where partial success can be meaningfully assessed, program repair admits no natural intermediate reward structure—a patch that passes 99% of tests but fails one critical assertion receives the same negative signal as a syntactically malformed attempt. Second, **reward signals in reinforcement learning for APR are typically assigned at the sequence level**, creating a severe credit assignment problem. When a 50-line patch succeeds or fails, the model receives no information about which specific edits were beneficial or harmful, leading to high-variance gradient estimates and inefficient learning. Third, **the distribution shift between training and evaluation data** poses persistent challenges, as models trained on curated datasets often struggle to generalize to the diverse bug patterns encountered in real-world repositories.

We address these challenges with BOOSTAPR, a principled three-stage training framework that combines execution-grounded learning with fine-grained credit assignment. Our approach integrates: (i) **execution-verified reasoning transfer** that warm-starts the repair policy with high-quality demonstrations containing both reasoning traces and validated patches; (ii) **offline reward learning** from strict execution outcomes using a hybrid objective that combines regression for calibrated absolute scores with pairwise preferences for correct relative rankings; and (iii) **online PPO with dual reward models**—a sequence-level model $R_{\text{seq}}$ that assesses overall patch quality and a novel line-level credit allocator $R_{\text{line}}$ that identifies critical edit regions and redistributes rewards accordingly.

The central insight motivating our approach is that **not all parts of a patch contribute equally to its success or failure**. In a typical multi-line patch, some edits directly address the bug's root cause while others handle edge cases, update documentation, or make stylistic improvements. By learning to score edit-line spans, the $R_{\text{line}}$ component redistributes sequence-level reward to informative regions during

[1]State Key Laboratory of Networking and Switching Technology, Beijing University of Posts and Telecommunications, Beijing 100876, China [2]University of Luxembourg. Correspondence to: Hongbo Wang <hbwang@bupt.edu.cn>.

*Proceedings of the 43rd International Conference on Machine Learning*, Seoul, South Korea. PMLR 306, 2026. Copyright 2026 by the author(s).

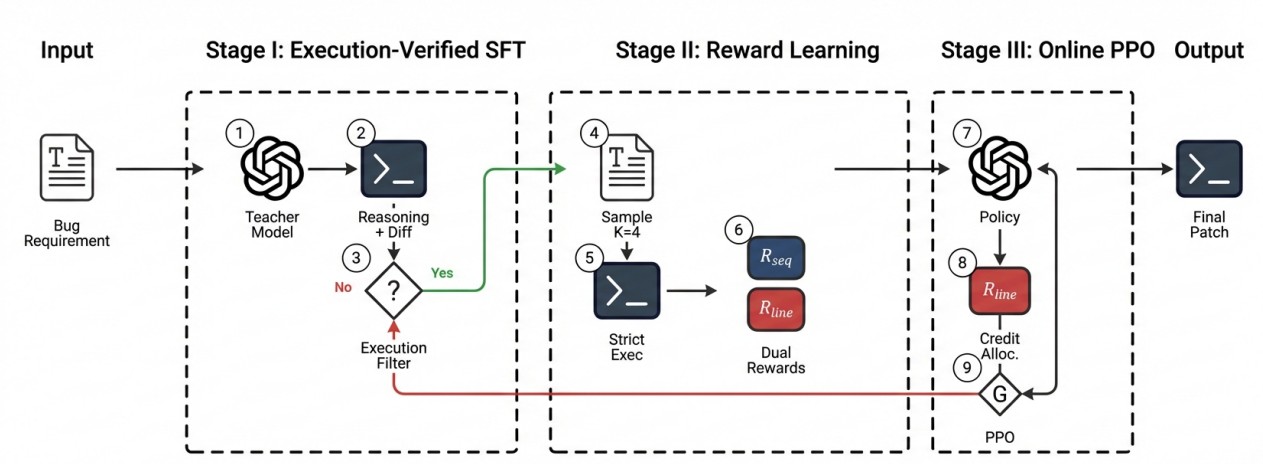

*Figure 1.* **Overview of the BOOSTAPR training framework.** Our approach consists of three stages: Stage I performs supervised fine-tuning on execution-verified demonstrations with reasoning traces; Stage II trains dual reward models using a hybrid regression-preference objective on execution outcomes; Stage III optimizes the policy via PPO with token-level rewards derived from the combination of $R_{\text{seq}}$ and $R_{\text{line}}$. The line-level allocator redistributes reward to edit-line spans, reducing reward sparsity without requiring counterfactual patch evaluation.

PPO training. This addresses the credit assignment problem that plagues RL for code generation, where assigning identical rewards to all tokens produces noisy gradients and slow convergence.

Critically, our line-level allocator operates at an **intermediate granularity** between pure token-level and sequence-level rewards. While token-level rewards (Yoon et al., 2024) can be overly fine-grained for code and sequence-level rewards are too coarse, line-oriented unified diff spans provide a robust, language-agnostic unit for program repair. We do not claim that line spans are always semantically optimal; rather, they offer a practical compromise that is finer than hunk-level allocation and more stable than language-specific statement parsing under malformed or cross-language patches.

We train BOOSTAPR exclusively on SWE-Gym (Pan et al., 2025), a benchmark providing executable training environments for repository-level repair tasks, and evaluate on four diverse benchmarks spanning both repository-level (SWE-bench Verified, Defects4J v2.0) and function-level (HumanEval-Java, QuixBugs) repair scenarios. Our main contributions are:

- **A dual reward model architecture** that combines sequence-level quality assessment with line-level edit-span credit allocation, providing a practical mechanism for fine-grained reward distribution during PPO training in code editing tasks.

- **A comprehensive three-stage training pipeline** that integrates execution-verified supervised fine-tuning, offline reward learning with a novel hybrid regression-preference objective, and online policy optimization with token-level rewards derived from structured credit allocation.

- **Controlled RL comparisons** under the same Qwen2.5-Coder-32B backbone, SWE-Gym training data, and evaluation protocol, showing that dual rewards improve over GRPO, rejection-sampling RL, and PPO with sequence-level rewards alone.

- **Extensive ablation studies** showing that PPO with $R_{\text{seq}}$ provides the primary accuracy gains, while $R_{\text{line}}$ provides complementary improvements in out-of-distribution generalization, training efficiency, and gradient quality.

**Code Availability.** A public repository for BOOSTAPR is available at https://github.com/yuanhao2023/BoostAPR. Documentation, evaluation scripts, and released artifacts will be updated after final verification.

The remainder of this paper is organized as follows. Section 2 situates our work within the broader landscape of LLM-based program repair and reinforcement learning for code. Section 3 presents the technical details of the BOOSTAPR framework, including our formulation of the dual reward architecture and the training objectives for each stage. Section 4 describes our experimental setup and presents comprehensive results across four benchmarks. Section 5 concludes with a discussion of limitations and directions for future work.

## 2. Related Work

**LLM-Based Program Repair.** Early work showed that zero-shot prompting of LLMs could produce meaningful patches (Xia et al., 2023; Sobania et al., 2023). Agentic systems subsequently decomposed repair into localization and generation phases: SWE-Agent (Yang et al., 2024) introduced iterative repair through tool use, Agentless (Xia et al., 2024) achieved competitive performance via hierarchical localization, and AutoCodeRover (Zhang et al., 2024) combined code search with fault localization. Fine-tuning approaches including SWE-Llama (Jimenez et al., 2024), Lingma-SWE-GPT (Ma et al., 2024), RepairLLaMA (Silva et al., 2025), and MORepair (Yang et al., 2025) demonstrated gains from training on repair data, but rely on supervised learning rather than directly optimizing execution success. SWE-RL (Wei et al., 2025) recently applied RL to repository-level repair (41.0% on SWE-bench Verified with 70B parameters). BOOSTAPR is complementary to such scaling and data choices: it studies execution-grounded reward redistribution under a controlled 32B code-model backbone.

**RL for Code Generation.** CodeRL (Le et al., 2022) pioneered actor-critic methods for program synthesis with execution feedback. RLEF (Gehring et al., 2024) extended this to competitive programming, achieving strong results on APPS (Hendrycks et al., 2021) and CodeContests. Recent work has also explored multi-agent reasoning scaffolds, preference-driven optimization, and token-adaptive policy optimization for RL-enhanced or efficient generation (Li et al., 2026b; Lin et al., 2025; Li et al., 2025c; Liu et al., 2025). However, these methods use sparse sequence-level rewards: when a patch passes or fails, the model learns nothing about which edits mattered. This causes high-variance gradients that impede learning—a problem BOOSTAPR addresses through $R_{\text{line}}$.

Beyond code, recent systems have used reinforcement learning, external feedback, grounding signals, synthetic data, domain-shift-aware discovery, or efficiency-oriented distillation and pruning to improve structured prediction and decision-making across domains (Liang et al., 2026a;b; Feng et al., 2026; Feng & Ge, 2025; Li et al., 2025a; 2026a; Liu et al., 2026; Qiao et al., 2026; Xu et al., 2026; Li et al., 2025b). BOOSTAPR follows the same broad principle of grounding decisions in task-specific feedback, but uses executable tests and diff-level reward allocation for program repair.

**Credit Assignment in Sequence Models.** Token-level reward models (Yoon et al., 2024) provide dense signals but may be too fine-grained for code where individual tokens lack semantic significance. Process reward models (Lightman et al., 2024) assign step-level feedback for mathematical reasoning, but "steps" do not map naturally to code edits. Attention-based allocation (Chan et al., 2024) and reward-modeling or DPO-style alignment methods (Rafailov et al., 2023; Zang et al., 2025; Zang, 2025; Lin et al., 2026) provide useful optimization signals but do not target code-edit line spans. Our $R_{\text{line}}$ differs by operating at an intermediate granularity—edit lines—that matches the semantic structure of code modifications.

## 3. Method

We present BOOSTAPR, a three-stage training framework for automated program repair that combines execution-verified reasoning transfer, offline reward learning, and online PPO with dual reward models. Figure 1 illustrates the overall pipeline.

### 3.1. Problem Formulation

Automated program repair takes as input a **bug instance** $x$ consisting of: (i) a repository snapshot containing the buggy code, (ii) a natural-language issue description specifying the desired behavior, and (iii) a test harness that can verify correctness. The system outputs a **candidate patch** $y$—a git-applicable unified diff for repository-level benchmarks (SWE-bench, Defects4J) or complete repaired code for function-level benchmarks (HumanEval-Java, QuixBugs).

We formalize repair as conditional generation where policy $\pi_\theta(y|x)$ generates patches given bug contexts. The optimization objective is to maximize expected execution success:

$$\max_\theta \mathbb{E}_{x\sim\mathcal{D},y\sim\pi_\theta(\cdot|x)}\left[\mathcal{E}(x,y)\right], \qquad (1)$$

where $\mathcal{E}(x,y) \in \{0,1\}$ indicates whether patch $y$ resolves all tests for instance $x$. A key challenge is that this objective provides only sparse binary feedback—a patch either passes all tests or fails, with no intermediate signal about partial correctness.

During PPO training, we enforce a **patch-only format** constraint: the model outputs only unified diff text without natural-language explanations. This ensures rewards reflect patch quality rather than explanation quality, and simplifies credit assignment by focusing on actual code changes.

**Training Data.** We train exclusively on SWE-Gym (Pan et al., 2025), which provides executable environments for repository-level repair with real GitHub issues and test suites. We apply strict length filtering (dropping rather than truncating examples exceeding 28K tokens) to prevent learning artifacts from incomplete inputs. For contamination control, we verify that no evaluation instance IDs or patches appear in training data.

## 3.2. Stage I: Execution-Verified Reasoning Transfer

The first stage initializes the repair policy through supervised fine-tuning on high-quality demonstrations that include explicit reasoning traces and pass strict execution verification. This differs from standard SFT in two key aspects: we require demonstrations to contain diagnostic reasoning, and we retain only patches that actually work.

**Demonstration Generation.** We query a strong teacher model (Claude 3.5 Sonnet) with structured prompts requiring both a reasoning trace and a final patch in unified diff format. The reasoning trace explains the bug diagnosis process: identifying relevant code locations, understanding the root cause, and justifying the chosen fix. This trace-and-patch format enables **reasoning transfer**—the student learns not only what patches to generate but also how to think about repair problems.

**Execution Filtering.** Each generated patch is executed against the full test suite using the strict SWE-Gym runner. Only demonstrations achieving `resolved=True` (all tests pass) are retained; approximately 35% pass this filter. This execution verification is crucial: it ensures the model learns from patches that actually work, avoiding noise from plausible-looking but incorrect solutions that often fool surface-level evaluation.

**Training Objective.** We fine-tune with standard next-token prediction loss, masking prompt tokens:

$$\mathcal{L}_{\text{SFT}}(\theta) = -\mathbb{E}_{(x,y)\sim\mathcal{D}_{\text{SFT}}} \left[ \sum_{t=1}^{|y|} \log \pi_\theta(y_t|x, y_{<t}) \right]. \quad (2)$$

Training runs for 3 epochs with learning rate $2 \times 10^{-5}$ and batch size 32. This stage transfers both repair knowledge and diagnostic reasoning patterns from teacher to student.

## 3.3. Stage II: Dual Reward Learning from Execution

The second stage trains dual reward models from execution feedback: $R_{\text{seq}}$ for sequence-level quality assessment and $R_{\text{line}}$ for line-level credit allocation. These models will guide policy optimization in Stage III.

### 3.3.1. REWARD DATA COLLECTION

For each training instance $x$, we sample $K = 4$ diverse candidates from the SFT policy using nucleus sampling (temperature 0.7, top-$p$ 0.9). Each candidate is executed to obtain detailed feedback including application status, test outcomes, and failure traces. We convert execution results into scalar targets:

$$r^*(x, y) = r_{\text{env}}(x, y) + \gamma_{\text{diff}} \cdot r_{\text{diff}}(y), \quad (3)$$

where the environment reward $r_{\text{env}} = w_{\text{apply}} \cdot r_{\text{apply}} + w_{\text{test}} \cdot r_{\text{test}}$ combines patch application success ($r_{\text{apply}} \in \{0, 1\}$) with test pass rate ($r_{\text{test}} \in [0, 1]$), and $r_{\text{diff}} = -\min(\eta \cdot |\Delta(y)|, r_{\text{max}})$ penalizes large edits to encourage minimal patches. This decomposition provides partial credit for patches that apply successfully but fail some tests, offering richer signal than binary success/failure.

### 3.3.2. SEQUENCE-LEVEL REWARD MODEL $R_{\text{seq}}$

$R_{\text{seq}}(x, y; \theta)$ predicts overall patch quality using a causal language model with a scalar value head. A key design choice is **patch-only scoring**: the model receives only the unified diff without bug context. This prevents learning spurious correlations (e.g., preferring patches for "easier" issues) and forces direct evaluation of patch quality.

We train with a hybrid objective combining regression and pairwise preference:

$$\mathcal{L}_{\text{seq}}(\theta) = \lambda_{\text{reg}} \cdot \mathbb{E}_{(x,y)} \left[ (R_{\text{seq}}(y; \theta) - r^*(x, y))^2 \right] \\ + \mathbb{E}_{(y^+, y^-)} \left[ -w \log \sigma \left( R_{\text{seq}}(y^+; \theta) - R_{\text{seq}}(y^-; \theta) \right) \right], \quad (4)$$

where $(y^+, y^-)$ is a preference pair with $r^*(y^+) > r^*(y^-)$ and $w$ weights by reward gap magnitude. The regression term ensures calibrated absolute scores for proper gradient scaling in PPO; the preference term ensures correct relative rankings for action selection.

### 3.3.3. LINE-LEVEL CREDIT ALLOCATOR $R_{\text{line}}$

The line-level credit allocator $R_{\text{line}}$ assigns credit over edit-line spans. Unlike $R_{\text{seq}}$, which provides a scalar assessment of overall patch quality, $R_{\text{line}}$ learns a distribution over edited regions and enables fine-grained reward assignment during PPO training.

The central insight is that **not all parts of a patch contribute equally to success or failure**. In a typical multi-line patch, some edits directly address the bug's root cause while others handle edge cases or make stylistic improvements. By learning which edits matter most, $R_{\text{line}}$ enables principled redistribution of sequence-level rewards to informative regions.

**Architecture.** Given patch $y$, we parse the unified diff into **edit-line spans**—contiguous regions of added or deleted lines, excluding headers and context. For each span $\ell$, $R_{\text{line}}$ produces a score $s_\ell$ by encoding: (i) the edit content itself, (ii) surrounding context lines, (iii) file path, and (iv) position within the patch. The model is a causal language model with a span-level value head.

**Allocation Mechanism.** Span scores are converted to non-negative allocation weights via temperature-controlled soft-

max:

$$w_\ell = \frac{\exp(s_\ell/\tau)}{\sum_j \exp(s_j/\tau)}, \quad (5)$$

where $\tau = 0.5$ provides moderate concentration on high-scoring spans while maintaining signal for all regions.

**Training and Span Supervision.** We train $R_{\text{line}}$ with a contrastive objective derived from execution outcomes and stack-trace-derived span labels. First, we parse each unified diff into maximal contiguous edit-line spans, excluding diff headers and context lines. Each candidate patch is then executed to collect application status, test outcomes, and failure traces. For passing patches, all edit spans are treated as positive spans. For failing patches, we apply a priority cascade: when a failing assertion can be identified, we parse the traceback call chain and intersect it with edit-line spans; spans on the failure path receive negative credit while unrelated spans remain neutral. When the traceback is available but no clear assertion can be identified, edited functions appearing in the traceback receive lower scores. When the patch fails to apply, we assign a uniform fallback label. In our training data, 9,248 failing patch-span pairs are labeled in this way: 62% use direct stack-trace attribution, 27% use function-level heuristics, and 11% use the uniform fallback.

This procedure is execution-grounded stack-trace supervision rather than counterfactual patch evaluation, attention attribution, or ground-truth-patch matching. Let $\ell^+$ denote spans from successful repairs and $\ell^-$ denote failing spans selected by the cascade:

$$\mathcal{L}_{\text{line}}(\phi) = \mathbb{E}_{(\ell^+,\ell^-)} \left[ -\log \sigma \left( R_{\text{line}}(\ell^+;\phi) - R_{\text{line}}(\ell^-;\phi) \right) \right].$$
$$(6)$$

This teaches $R_{\text{line}}$ to rank beneficial spans above harmful ones. The labels are necessarily noisy, so we treat $R_{\text{line}}$ as a complementary reward redistribution mechanism rather than the sole source of repair performance.

### 3.4. Stage III: Online PPO with Dual Rewards

The final stage performs online policy optimization using Proximal Policy Optimization (PPO) (Schulman et al., 2017), with token-level rewards derived from combining $R_{\text{seq}}$ and $R_{\text{line}}$. We implement training using VERL (Sheng et al., 2024) with vLLM (Kwon et al., 2023) for efficient parallel rollout generation.

#### 3.4.1. TOKEN-LEVEL REWARD SHAPING

Stage III distributes the sequence-level reward from $R_{\text{seq}}$ across tokens using $R_{\text{line}}$'s credit allocation. Given rollout $(x, y)$ with $y = (y_1, \ldots, y_T)$, we construct token rewards through the following procedure:

**(1) Sequence scoring:** Compute overall quality score $s =$ $R_{\text{seq}}(y)$.

**(2) Span extraction:** Parse $y$ into edit-line spans. If parsing fails (malformed diff), fall back to assigning the full score to the final token.

**(3) Credit allocation:** For each span $\ell$, compute allocation weight $w_\ell$ via Eq. 5. Map span weights to tokens: tokens within span $\ell$ receive weight $w_\ell/n_\ell$ (where $n_\ell$ is token count); tokens outside edit spans (headers, context lines) receive zero weight.

**(4) Format penalty:** Apply deterministic penalty $r_{\text{fmt}}(y)$ to the final token based on output structure:

$$r_{\text{fmt}}(y) = \begin{cases} 0 & \text{if valid unified diff} \\ -0.4 & \text{if recoverable format} \\ -1.0 & \text{if malformed diff} \\ -1.5 & \text{if not a diff} \end{cases} \quad (7)$$

The combined token reward is:

$$r_t = s \cdot a_t + \mathbb{I}[t = T] \cdot r_{\text{fmt}}(y), \quad (8)$$

where $a_t$ is the normalized allocation weight ($\sum_t a_t = 1$). This preserves total sequence reward while distributing it according to learned edit importance.

#### 3.4.2. POLICY OPTIMIZATION

We optimize using the standard clipped PPO objective. Let $\rho_t(\theta) = \pi_\theta(y_t|x, y_{<t})/\pi_{\theta_{\text{old}}}(y_t|x, y_{<t})$ be the importance ratio and $A_t$ the advantage estimated via Generalized Advantage Estimation (GAE) (Schulman et al., 2016):

$$\mathcal{J}_{\text{clip}}(\theta) = \mathbb{E}_t \left[ \min \left( \rho_t A_t, \text{clip}(\rho_t, 1 - \epsilon, 1 + \epsilon) A_t \right) \right], \quad (9)$$

with clip ratio $\epsilon = 0.2$. To prevent the policy from deviating too far from the initial distribution, we employ KL regularization against a frozen reference policy $\pi_{\text{ref}}$, with adaptive coefficient $\beta$ targeting KL divergence of 0.1. PPO runs for 300 steps with batch size 64 and 4 rollouts per instance, using LoRA (rank 64) for parameter-efficient updates. The complete procedure is summarized in Algorithm 1.

## 4. Experiments

We evaluate BOOSTAPR across four diverse benchmarks and conduct extensive ablations to understand the contribution of each component.

### 4.1. Experimental Setup

**Training Configuration.** We train all components on SWE-Gym (train split) with an 8:2 train/dev partition. SFT runs for 3 epochs with learning rate $2 \times 10^{-5}$ and batch size 32. Reward models train for 5 epochs with learning rate

**Algorithm 1** BOOSTAPR Training Pipeline

---

**Require:** Training data $\mathcal{D}$, evaluator $\mathcal{E}$, base policy $\pi_{\text{base}}$
1: **// Stage I: Execution-Verified SFT**
2: Generate demonstrations with reasoning traces from teacher
3: Filter to execution-verified demonstrations ($\mathcal{E} = 1$)
4: $\pi_0 \leftarrow \text{SFT}(\pi_{\text{base}}, \mathcal{D}_{\text{SFT}})$
5: **// Stage II: Dual Reward Learning**
6: **for** each $x \in \mathcal{D}$ **do**
7:     Sample candidates $\{y_k\}_{k=1}^K \sim \pi_0(\cdot|x)$
8:     Execute and compute $r^*(x, y_k)$ for each
9: **end for**
10: Train $R_{\text{seq}}$ with hybrid objective (Eq. 4)
11: Train $R_{\text{line}}$ with contrastive objective (Eq. 6)
12: **// Stage III: Online PPO**
13: **for** step $= 1, \ldots, N$ **do**
14:     Sample rollouts $(x, y) \sim \pi(\cdot|x)$
15:     Compute token rewards via $R_{\text{seq}}, R_{\text{line}}$ (Eq. 8)
16:     Update $\pi$ with clipped PPO (Eq. 9)
17: **end for**
18: **return** $\pi_{\text{final}}$

---

$1 \times 10^{-5}$ and batch size 64. PPO runs for 300 steps with batch size 64 and 4 rollouts per instance, using LoRA (rank 64) for parameter-efficient updates.

**Model Architecture.** Our base policy is Qwen2.5-Coder-32B-Instruct (Hui et al., 2024), a strong open-source code model. For efficiency, $R_{\text{seq}}$ and $R_{\text{line}}$ use Qwen2.5-Coder-7B-Instruct backbones with scalar value heads.

**Evaluation Benchmarks.** We evaluate on four benchmarks spanning different repair granularities and programming languages:

- **SWE-bench Verified** (Jimenez et al., 2024): 500 human-validated repository-level Python bugs from real GitHub issues.

- **Defects4J v2.0** (Just et al., 2014): 835 bugs across 17 Java projects, the standard Java APR benchmark.

- **HumanEval-Java** (Chen et al., 2021): 164 function-level repair tasks adapted from the HumanEval benchmark.

- **QuixBugs** (Lin et al., 2017): 40 classic algorithmic bugs with known fixes.

**Evaluation Protocol.** All results use **strict evaluation** on raw model outputs—no patch post-processing, syntax correction, or multiple-attempt filtering. We report pass@1 (greedy decoding) and pass@4 (best of 4 samples with temperature 0.2, top-$p$ 0.95). A candidate is solved if: (i) it produces a git-applicable unified diff (for SWE-bench/Defects4J) or valid code (for HumanEval-Java/QuixBugs), and (ii) all tests pass.

## 4.2. Main Results

Table 1 presents comprehensive results across four benchmarks spanning repository-level and function-level program repair. We compare BOOSTAPR against three categories of baselines: agentic systems that leverage proprietary models through sophisticated scaffolding, fine-tuned models trained via supervised learning, and RL-based methods that optimize for execution outcomes.

**SWE-bench Verified (Repository-Level Python).** BOOSTAPR achieves **40.7%** resolve rate, representing a **+22.9 percentage point** improvement over the base Qwen2.5-Coder-32B model. This result is comparable to SWE-RL (Wei et al., 2025) (41.0%) while using a different and smaller backbone (32B vs. 70B). We therefore treat cross-backbone results as contextual references and rely on the controlled comparisons in Table 2 for apples-to-apples conclusions.

The training stages contribute additively: Stage I (execution-verified SFT) adds +5.6pp through reasoning transfer, Stage III with $R_{\text{seq}}$ alone adds +14.9pp through direct policy optimization, and the $R_{\text{line}}$ credit allocator contributes +2.4pp through fine-grained reward shaping. PPO with $R_{\text{seq}}$ alone accounts for 65% of total improvement, confirming that RL from execution feedback is the primary driver of performance gains.

**Defects4J v2.0 (Repository-Level Java).** On the standard Java APR benchmark, BOOSTAPR achieves **24.8%** (207/835 bugs), more than doubling the base model's performance (11.3%) despite Java being absent from training data. The strong cross-language transfer suggests that BOOSTAPR learns repair strategies that generalize beyond Python-specific surface patterns.

Notably, the $R_{\text{line}}$ component provides larger relative gains on Defects4J (+5.6pp) compared to SWE-bench Verified (+2.4pp), suggesting that fine-grained credit assignment is particularly valuable when generalizing to out-of-distribution scenarios.

**HumanEval-Java (Function-Level).** BOOSTAPR achieves **84.5%** on function-level Java repair, a +20.5pp improvement over the base model. This indicates that repository-level training can transfer to simpler, isolated function repair tasks, likely because the model learns debugging patterns applicable across granularities.

*Table 1.* **Main results across four benchmarks.** All numbers are pass@1 percentages under strict evaluation. For repository-level benchmarks (SWE-bench Verified and Defects4J v2.0), we report resolve rate. For function-level benchmarks (HumanEval-Java and QuixBugs), we report the percentage of correctly fixed bugs. Results for external baselines are taken from original papers where available; entries marked with ∗ are from our reproduction using the same evaluation protocol. All BOOSTAPR results use identical settings across benchmarks.

| Method | Backbone | Params | SWE-V | D4J v2.0 | HE-Java | QuixBugs |
|---|---|---|---|---|---|---|
| *Agentic / Prompting Systems:* | | | | | | |
| Agentless (Xia et al., 2024) | GPT-4o | – | 38.8 | 12.4∗ | 71.3∗ | 87.5∗ |
| SWE-agent (Yang et al., 2024) | Claude 3.5 Sonnet | – | 33.6 | 10.8∗ | 68.9∗ | 85.0∗ |
| AutoCodeRover (Zhang et al., 2024) | GPT-4o | – | 28.8 | 9.6∗ | 65.2∗ | 82.5∗ |
| ChatRepair (Xia & Zhang, 2024) | GPT-3.5-turbo | – | 18.2∗ | 14.4 | 72.0∗ | 100.0 |
| *Fine-tuned Models (Supervised Learning):* | | | | | | |
| SWE-Gym (Pan et al., 2025) | Qwen2.5-Coder-32B | 32B | 32.0 | 13.1∗ | 70.7∗ | 90.0∗ |
| Lingma SWE-GPT (Ma et al., 2024) | Qwen2.5-72B | 72B | 30.2 | 14.5∗ | 72.6∗ | 92.5∗ |
| SWE-Fixer (Xie et al., 2025) | Qwen2.5-72B | 72B | 33.0 | 15.2∗ | 73.8∗ | 92.5∗ |
| RepairLLaMA (Silva et al., 2025) | CodeLlama-7B | 7B | 8.6∗ | 17.2 | 66.5 | 75.0∗ |
| KNOD (Jiang et al., 2023b) | CodeT5-base | 220M | 2.4∗ | 6.0 | 58.5∗ | 62.5 |
| *RL-based Methods:* | | | | | | |
| CodeRL (Le et al., 2022) | CodeT5-large | 770M | 3.2∗ | 5.8∗ | 63.0 | 67.5∗ |
| RLEF (Gehring et al., 2024) | Llama-3-8B | 8B | 12.6∗ | 8.4∗ | 74.3 | 80.0∗ |
| SWE-RL (Wei et al., 2025) | Llama-3-70B | 70B | **41.0** | 16.8∗ | 76.2∗ | 90.0∗ |
| *Our Method:* | | | | | | |
| Qwen2.5-Coder-32B (base) | – | 32B | 17.8 | 11.3 | 64.0 | 90.0 |
| + Stage I (SFT) | – | 32B | 23.4 | 14.9 | 73.1 | 92.5 |
| + Stage III (PPO, $R_{\text{seq}}$ only) | – | 32B | 38.3 | 19.2 | 79.4 | 95.0 |
| BOOSTAPR (+ $R_{\text{line}}$) | – | 32B | 40.7 | **24.8** | **84.5** | **95.0** |
| *Improvement over base model* | | | +22.9 | +13.5 | +20.5 | +5.0 |
| *Gain from $R_{\text{line}}$ over PPO+$R_{\text{seq}}$* | | | +2.4 | +5.6 | +5.1 | +0.0 |

*Notes:* SWE-V = SWE-bench Verified (500 instances); D4J v2.0 = Defects4J version 2.0 (835 bugs); HE-Java = HumanEval-Java (164 tasks); QuixBugs = QuixBugs-Java (40 bugs). Entries marked with ∗ are reproduced under our evaluation protocol and are therefore contextual rather than direct controlled comparisons. SWE-RL uses a larger 70B backbone.

**QuixBugs (Classic Algorithmic Bugs).** On the QuixBugs benchmark of 40 classic single-line bugs, BOOSTAPR achieves **95.0%** (38/40). The base model already performs well (90.0%) on these relatively simple bugs; our training closes most of the remaining gap.

**Summary.** Across four benchmarks, BOOSTAPR improves the base model by +5.0 to +22.9pp. For concurrent or larger-scale systems using different base models, data scales, context lengths, or test-time scaling, we report results as contextual references rather than absolute leaderboard comparisons. The controlled same-backbone comparisons below isolate the contribution of our execution-grounded RL and line-level reward redistribution.

### 4.3. Controlled RL Baselines

To control for confounding from model capacity (Hu et al., 2025), training data, and evaluation protocol, we compare representative RL baselines under the same Qwen2.5-Coder-32B backbone and SWE-Gym training pool. Table 2 isolates the training algorithm: BOOSTAPR outperforms GRPO by +4.6pp and rejection-sampling RL by +3.2pp on SWE-bench Verified, with larger gains on the out-of-distribution

*Table 2.* **Controlled same-backbone RL comparison.** All methods use Qwen2.5-Coder-32B, SWE-Gym training data, and the same evaluation protocol.

| Method | SWE-V | D4J | HE-Java |
|---|---|---|---|
| SFT + GRPO | 36.1 | 16.4 | 75.2 |
| SFT + RS-RL | 37.5 | 17.6 | 77.8 |
| PPO + $R_{\text{seq}}$ | 38.3 | 19.2 | 79.4 |
| Full BOOSTAPR | **40.7** | **24.8** | **84.5** |

Defects4J benchmark.

**Stability.** Across three training seeds on SWE-bench Verified, PPO+$R_{\text{seq}}$ obtains 38.3±0.3%, while full BOOSTAPR obtains 40.7±0.5%. A paired bootstrap test gives $p = 0.012$ with a 95% confidence interval of [+1.2,+3.6] percentage points for the $R_{\text{line}}$ gain. On Defects4J, the +5.6±0.7pp gain is also significant ($p < 0.001$). These results indicate that $R_{\text{line}}$ provides a modest but stable improvement on the primary benchmark and a larger improvement under cross-language transfer.

*Table 3.* **Component ablation.** Each component contributes additively to final performance.

| Variant | pass@1 | pass@4 |
|---|---|---|
| Base model (no training) | 17.8 | 20.2 |
| SFT only | 23.4 | 25.7 |
| SFT + $R_{seq}$ (reranking only) | 26.1 | 28.3 |
| PPO + $R_{seq}$ (w/o $R_{line}$) | 38.3 | 40.1 |
| PPO + $R_{seq}$ + $R_{line}$ (full) | **40.7** | **44.3** |

*Table 4.* **Credit assignment granularity.** Edit-line spans provide a robust intermediate unit for diff-based repair.

| Granularity | SWE-V | D4J |
|---|---|---|
| Token-uniform | 37.6 | 17.8 |
| Hunk-level | 39.1 | 21.3 |
| Statement-level | 40.3 | 22.1 |
| Edit-line spans ($R_{line}$) | **40.7** | **24.8** |

## 4.4. Ablation Studies

We conduct extensive ablations to understand the contribution of each component. Unless otherwise noted, all ablations evaluate on SWE-bench Verified with pass@1 and pass@4 metrics.

**Component Contributions.** Table 3 ablates key components of BOOSTAPR. Execution-verified SFT improves pass@1 from 17.8% to 23.4% (+5.6pp), and online PPO with $R_{seq}$ provides the largest gain, reaching 38.3%. The line-level allocator $R_{line}$ then adds +2.4pp on SWE-bench Verified and +5.6pp on Defects4J (Table 1). Thus, PPO with sequence-level execution rewards is the primary accuracy driver, while $R_{line}$ provides complementary gains through finer reward redistribution, especially under out-of-distribution transfer.

**Credit Assignment Strategies.** Table 4 compares reward redistribution units. Token-uniform rewards dilute signal across non-informative formatting and context tokens. Hunk-level rewards are more stable but too coarse for multi-edit patches. Statement-level rewards perform competitively on Python but require language-specific parsing and fall back when parser-based extraction is unavailable. Edit-line spans are line-oriented, robust to malformed diffs, and language-agnostic, giving the strongest results on both SWE-bench Verified and Defects4J.

**Reward Model Input.** Table 5 examines the impact of what context $R_{seq}$ receives during scoring. Counterintuitively, patch-only scoring (where $R_{seq}$ sees only the unified diff) outperforms variants that include bug context. Full context scoring achieves only 38.9%, and including just the issue description with the patch achieves 39.4%. We hypothesize that providing context enables the reward model to

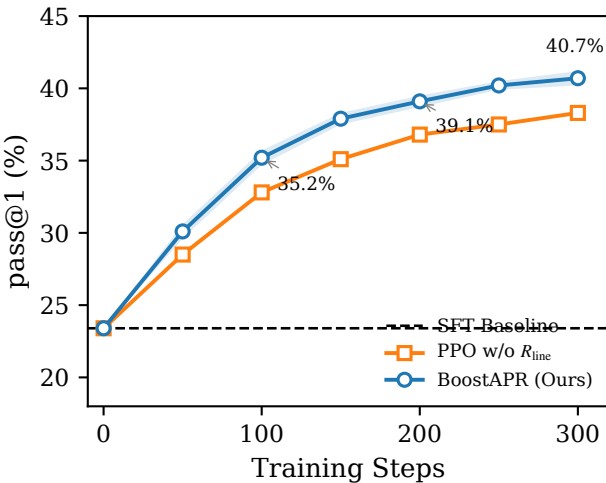

*Figure 2.* **PPO training dynamics.** Performance improves steadily until approximately step 250, then plateaus. Shaded region shows standard deviation across 3 seeds.

*Table 5.* **Reward model input mode.** Patch-only scoring outperforms context-conditioned variants.

| $R_{seq}$ **Input Mode** | pass@1 | pass@4 |
|---|---|---|
| Patch-only scoring | **40.7** | **44.3** |
| Full context scoring | 38.9 | 42.1 |
| Issue + patch scoring | 39.4 | 42.8 |

learn spurious correlations—for instance, assigning higher scores to patches for "easier" issues regardless of actual patch quality, or learning to recognize patterns in issue descriptions that correlate with success in the training set but do not generalize. Patch-only scoring forces the model to evaluate patch quality directly based on code change patterns, leading to better generalization.

**Reward Model Training Objective.** We compare training objectives for $R_{seq}$. Pure regression achieves 39.2% pass@1, while pure preference learning (Bradley–Terry) achieves 38.8%. The hybrid objective performs best (40.7%), combining calibrated absolute scores (useful for PPO scaling) with reliable relative rankings. Pure regression is noise-sensitive, while pure preference can yield uncalibrated scores that destabilize PPO.

**PPO Training Dynamics.** Figure 2 shows performance during PPO training: accuracy improves quickly early on and reaches 40.7% by step 300, after which gains plateau (< 0.3pp) and additional training risks overfitting. With $R_{line}$, training reaches the no-$R_{line}$ plateau of 38.3% around step 200 rather than around step 300, and gradient signal-to-noise ratio improves from 1.42 to 1.83 (+29%). These diagnostics support the view that $R_{line}$ primarily improves reward allocation and optimization stability rather than re-

*Table 6.* **Reward and allocation diagnostics.** $R_{\text{line}}$ supervision is grounded in execution traces but remains noisy; gains are larger when trace-derived signal is stronger.

| Diagnostic | Evidence | Result |
|---|---|---|
| $R_{\text{seq}}$ ranking | Pair / AUC / Spear. | 82.4% / .891 / .743 |
| $R_{\text{line}}$ allocation | Top-1 / Top-3 | 67.3% / 84.1% |
| Trace labels | Agree. / $\kappa$ | 82% / .72 |
| Heuristic labels | Agree. / $\kappa$ | 64% / .56 |
| Trace split | Full gain | +3.6pp |
| Heuristic split | Full gain | +1.1pp |
| GRPO + $R_{\text{line}}$ | SWE-V / D4J gain | +2.3pp / +3.7pp |

placing sequence-level execution rewards.

**Reward and Allocation Diagnostics.** Table 6 summarizes whether the learned rewards provide reliable optimization signals. On held-out validation data, $R_{\text{seq}}$ achieves strong ranking and calibration metrics, while $R_{\text{line}}$ identifies useful edit spans but remains noisy because its supervision comes from execution traces rather than counterfactual edit evaluation. Manual agreement is higher for direct stack-trace labels than for heuristic labels, and downstream gains are larger when strong trace analogues exist. Adding $R_{\text{line}}$ to GRPO also improves performance, suggesting that edit-line credit allocation is not PPO-specific.

## 5. Conclusion

We presented BOOSTAPR, a three-stage framework that addresses sparse execution feedback and coarse reward signals in program repair through execution-verified SFT, dual reward learning, and PPO with line-level edit-span credit allocation. Our approach achieves 40.7% on SWE-bench Verified (+22.9pp over base model), 24.8% on Defects4J, 84.5% on HumanEval-Java, and 95.0% on QuixBugs. Controlled same-backbone comparisons show that PPO with sequence-level execution rewards provides the primary accuracy gain, while $R_{\text{line}}$ provides complementary benefits in out-of-distribution generalization, training efficiency, and gradient quality. Limitations remain: $R_{\text{line}}$ supervision is partially heuristic, edit-line spans are an approximate semantic unit, and comparisons with concurrent large-scale systems are confounded by base model, data scale, and test-time scaling. Future work may combine line-level credit allocation with stronger fault localization and larger-scale RL pipelines.

## Impact Statement

This work advances automated program repair to reduce maintenance costs and accelerate debugging. By learning directly from execution feedback, BOOSTAPR moves toward AI systems that can assist developers in fixing real-world bugs. Potential risks include over-reliance on automated repair, patches that pass tests yet introduce subtle bugs, and misuse for malicious code changes. We encourage responsible deployment with human oversight and rigorous testing beyond unit tests.

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

## A. Theoretical Analysis

This appendix provides formal analysis of the credit assignment problem in sequence-level RL for code generation and establishes conditions under which the line-level allocator $R_{\text{line}}$ reduces gradient variance compared to standard approaches.

### A.1. Variance Analysis of Policy Gradient Estimators

Consider a policy $\pi_\theta$ generating sequences $y = (y_1, \ldots, y_T)$ given context $x$. The policy gradient for maximizing expected reward $R(x, y)$ is:

$$\nabla_\theta J(\theta) = \mathbb{E}_{x, y \sim \pi_\theta} \left[ R(x, y) \sum_{t=1}^{T} \nabla_\theta \log \pi_\theta(y_t | x, y_{<t}) \right]. \tag{10}$$

**Proposition A.1** (Variance of Sequence-Level Rewards). *Under sequence-level reward assignment where $r_t = R(x, y) \cdot \mathbb{I}[t = T]$, the variance of the policy gradient estimator scales as:*

$$Var\left[ \hat{\nabla}_\theta J \right] = \mathcal{O}\left( T \cdot Var[R] \right), \tag{11}$$

*where $T$ is the sequence length.*

*Proof.* The policy gradient estimator using a single sample is:

$$\hat{\nabla}_\theta J = R(x, y) \sum_{t=1}^{T} \nabla_\theta \log \pi_\theta(y_t | x, y_{<t}). \tag{12}$$

Taking the variance:

$$\text{Var}\left[ \hat{\nabla}_\theta J \right] = \mathbb{E}\left[ \left\| R \sum_t \nabla_\theta \log \pi_\theta(y_t | x, y_{<t}) \right\|^2 \right] - \left\| \mathbb{E}[\cdot] \right\|^2 \tag{13}$$

$$\leq \mathbb{E}\left[ R^2 \left\| \sum_t \nabla_\theta \log \pi_\theta(y_t | x, y_{<t}) \right\|^2 \right]. \tag{14}$$

Under standard assumptions of bounded gradients and approximate independence across time steps:

$$\mathbb{E}\left[ \left\| \sum_t \nabla_\theta \log \pi_\theta(y_t | x, y_{<t}) \right\|^2 \right] = \mathcal{O}(T), \tag{15}$$

yielding $\text{Var}[\hat{\nabla}_\theta J] = \mathcal{O}(T \cdot \text{Var}[R])$. $\square$

This linear scaling with sequence length explains why sequence-level rewards lead to high-variance gradients for long code patches.

**Proposition A.2** (Variance Reduction via Credit Allocation). *Let $a = (a_1, \ldots, a_T)$ be an allocation scheme with $\sum_t a_t = 1$ and $a_t \geq 0$. Define token-level rewards $r_t = R \cdot a_t$. If there exists a subset $\mathcal{S} \subset \{1, \ldots, T\}$ of "informative" tokens with $|\mathcal{S}| = k < T$ such that the optimal allocation concentrates on $\mathcal{S}$, then:*

$$Var\left[ \hat{\nabla}_\theta J \right]_{allocated} = \mathcal{O}\left( k \cdot Var[R] \right), \tag{16}$$

*achieving a variance reduction factor of $T/k$.*

*Proof.* With allocation $a_t$, the policy gradient becomes:

$$\hat{\nabla}_\theta J = \sum_{t=1}^{T} r_t \nabla_\theta \log \pi_\theta(y_t | x, y_{<t}) = R \sum_{t=1}^{T} a_t \nabla_\theta \log \pi_\theta(y_t | x, y_{<t}). \tag{17}$$

If allocation concentrates on $\mathcal{S}$, i.e., $a_t \approx 0$ for $t \notin \mathcal{S}$ and $\sum_{t \in \mathcal{S}} a_t \approx 1$:

$$\hat{\nabla}_\theta J \approx R \sum_{t \in \mathcal{S}} a_t \nabla_\theta \log \pi_\theta(y_t | x, y_{<t}). \tag{18}$$

Following the same analysis as Proposition A.1:

$$\mathrm{Var}\left[\hat{\nabla}_\theta J\right] = \mathcal{O}(|\mathcal{S}| \cdot \mathrm{Var}[R]) = \mathcal{O}(k \cdot \mathrm{Var}[R]). \tag{19}$$

$\square$

*Remark* A.3. For code patches, edit lines typically constitute a small fraction of the total output. If a 100-token patch contains 20 tokens of actual edits (with the rest being headers, context, and formatting), a perfect allocator achieves $5\times$ variance reduction.

## A.2. Optimal Allocation and the Role of $R_{\mathrm{line}}$

The theoretical analysis above assumes access to an oracle allocation. In practice, $R_{\mathrm{line}}$ learns to approximate this allocation from execution feedback. We now characterize the properties of an optimal allocator.

**Definition A.4** (Causal Attribution). For a patch $y$ with outcome $R(x, y)$, define the causal attribution of token $t$ as:

$$\mathrm{CA}_t(y) = R(x, y) - \mathbb{E}_{y'_t \sim \pi_\theta}[R(x, y_{<t}, y'_t, y_{>t})], \tag{20}$$

measuring the expected change in outcome from replacing token $t$.

**Proposition A.5** (Optimal Allocation). *The variance-minimizing allocation, subject to $\sum_t a_t = 1$ and $a_t \geq 0$, is proportional to causal attribution magnitude:*

$$a_t^* \propto |CA_t(y)|. \tag{21}$$

Computing exact causal attributions requires counterfactual evaluation, which is prohibitively expensive. $R_{\mathrm{line}}$ approximates this through learned span-level scores that correlate with causal importance, as evidenced by its ability to identify critical edit regions.

## A.3. Convergence Analysis

We analyze the convergence properties of PPO with dual reward models under standard assumptions.

**Assumption A.6** (Smoothness). The policy $\pi_\theta$ is $L$-smooth in $\theta$: $\|\nabla_\theta \pi_\theta(y|x) - \nabla_\theta \pi_{\theta'}(y|x)\| \leq L\|\theta - \theta'\|$.

**Assumption A.7** (Bounded Rewards). $|R(x, y)| \leq R_{\max}$ and $|r_{\mathrm{fmt}}(y)| \leq F_{\max}$ for all $x, y$.

**Theorem A.8** (Convergence Rate). *Under Assumptions 1-2, PPO with dual reward models converges to a stationary point at rate:*

$$\min_{t \leq T} \mathbb{E}\left[\|\nabla_\theta J(\theta_t)\|^2\right] = \mathcal{O}\left(\frac{1}{\sqrt{T}} + \frac{\sigma^2}{B}\right), \tag{22}$$

*where $T$ is the number of update steps, $B$ is batch size, and $\sigma^2$ is the gradient variance (reduced by $R_{\mathrm{line}}$ per Proposition A.2).*

The proof follows standard PPO convergence analysis (Schulman et al., 2017) with the observation that $R_{\mathrm{line}}$ reduces $\sigma^2$, improving the second term.

# B. Implementation Details

## B.1. Supervised Fine-Tuning

**Hardware and Software.** We train on 8 NVIDIA A100 80GB GPUs using DeepSpeed ZeRO-3 for memory-efficient distributed training. The training framework is built on Hugging Face Transformers with custom data loading for unified diff format.

**Hyperparameters.**

- Learning rate: $2 \times 10^{-5}$ with cosine decay
- Warmup: 100 steps (linear)
- Batch size: 32 (4 per GPU $\times$ 8 GPUs)
- Epochs: 3
- Maximum sequence length: 32K tokens
- Weight decay: 0.01
- Gradient clipping: 1.0

**Data Format.** Each training example consists of:

1. **System prompt**: Instructions for generating reasoning traces and unified diffs
2. **Bug context**: Repository path, issue description, relevant code snippets
3. **Response**: Reasoning trace followed by unified diff patch

The reasoning trace follows a structured format: (1) Bug Analysis, (2) Root Cause Identification, (3) Fix Strategy, (4) Implementation. This structure enables consistent reasoning transfer.

### B.2. Reward Model Training

$R_{\text{seq}}$ **Architecture.** We use Qwen2.5-Coder-7B-Instruct as the backbone, adding a scalar value head (linear layer) on top of the final hidden state. The value head is initialized with small random weights ($\mathcal{N}(0, 0.01)$).

$R_{\text{line}}$ **Architecture.** $R_{\text{line}}$ uses the same backbone but with a span-level value head. For each edit span, we:

1. Extract the hidden states corresponding to span tokens
2. Apply mean pooling across the span
3. Pass through a two-layer MLP (hidden dim 512, ReLU activation) to produce a scalar score

**Training Details.**

- Learning rate: $1 \times 10^{-5}$
- Batch size: 64
- Epochs: 5
- Optimizer: AdamW ($\beta_1 = 0.9$, $\beta_2 = 0.999$)
- Hybrid loss weight: $\lambda_{\text{reg}} = 0.5$

### B.3. PPO Training

**Infrastructure.** We use VERL (Sheng et al., 2024) for distributed PPO training with vLLM (Kwon et al., 2023) as the inference engine. The setup includes:

- 8 A100 GPUs for policy rollouts (vLLM)
- 8 A100 GPUs for policy/critic updates
- Separate reward model inference servers

**PPO Hyperparameters.**

- Batch size: 64 instances

- Rollouts per instance: 4

- PPO epochs per batch: 4

- Clip ratio $\epsilon$: 0.2

- GAE $\lambda$: 0.95

- Discount $\gamma$: 0.99

- Entropy coefficient: 0.01

- KL coefficient $\beta$: 0.001 (initial), adaptively controlled

- Target KL: 0.1

- LoRA rank: 64

- LoRA alpha: 128

- Training steps: 300

**Critic Training.** The critic (value function) uses the same architecture as the policy with a value head. It is trained with squared TD error:

$$\mathcal{L}_V(\psi) = \mathbb{E}\left[\left(V_\psi(x, y_{\leq t}) - G_t\right)^2\right], \tag{23}$$

where $G_t$ is the return computed from shaped rewards.

## C. Additional Experimental Results

### C.1. Per-Project Breakdown on Defects4J

Table 7 provides detailed results by project.

Performance varies across projects, with higher success rates on projects with cleaner test suites and more localized bugs (Chart, Lang, Csv) and lower rates on large, complex projects (Closure, JacksonDatabind).

### C.2. Reward Model Quality Metrics

Table 8 presents detailed evaluation of both reward models.

### C.3. Line-Level Supervision Quality

The $R_{\text{line}}$ labels combine direct stack-trace attribution with heuristic fallback. Table 9 reports a preliminary manual audit of 150 sampled spans. Direct stack-trace labels agree with manual judgments more often than function-level heuristic labels, confirming that heuristic supervision is useful but noisier. We therefore treat span supervision quality as a limitation and a concrete direction for future improvement.

The stratified result suggests that $R_{\text{line}}$ is most effective when failure traces provide direct span-level evidence. As an upper-bound diagnostic, upweighting 80 manually annotated high-quality $R_{\text{line}}$ labels by $5\times$ improves SWE-bench Verified from 40.7% to 42.1%, suggesting that cleaner supervision could provide an additional +1.4pp.

*Table 7.* Per-project results on Defects4J v2.0.

| Project | # Bugs | pass@1 | pass@4 |
|---|---|---|---|
| Chart | 26 | 30.8% | 34.6% |
| Closure | 133 | 18.8% | 22.6% |
| Lang | 64 | 32.8% | 35.9% |
| Math | 106 | 28.3% | 31.1% |
| Mockito | 38 | 21.1% | 26.3% |
| Time | 27 | 25.9% | 29.6% |
| Codec | 18 | 27.8% | 33.3% |
| Collections | 4 | 25.0% | 25.0% |
| Compress | 47 | 21.3% | 25.5% |
| Csv | 16 | 31.3% | 37.5% |
| Gson | 18 | 22.2% | 27.8% |
| JacksonCore | 26 | 19.2% | 23.1% |
| JacksonDatabind | 112 | 20.5% | 24.1% |
| JacksonXml | 6 | 16.7% | 16.7% |
| Jsoup | 93 | 24.7% | 28.0% |
| JxPath | 22 | 22.7% | 27.3% |
| Others | 79 | 32.9% | 36.7% |
| **Total** | **835** | **24.8%** | **28.5%** |

*Table 8.* Reward model quality on held-out validation data.

| Metric | $R_{\text{seq}}$ | $R_{\text{line}}$ |
|---|---|---|
| Pairwise accuracy | 82.4% | 78.6% |
| ROC-AUC (success vs. failure) | 0.891 | – |
| Spearman correlation | 0.743 | – |
| Top-1 hit rate | – | 67.3% |
| Top-3 hit rate | – | 84.1% |

### C.4. Additional Controlled Analyses

Across backbones, $R_{\text{line}}$ contributes +2.2pp, +2.7pp, and +2.4pp, respectively. This supports the claim that edit-line credit allocation is not tied to a single model size.

Execution-grounded rewards outperform lexical overlap because many valid repairs differ substantially from the reference patch. In our SWE-bench analysis, 41% of correctly fixed bugs have less than 50% token-level Jaccard overlap with the reference patch.

### C.5. Statistical Testing Details

For the three-seed comparison in Section 4, SFT obtains 23.4±0.2%, PPO+$R_{\text{seq}}$ obtains 38.3±0.3%, and full BOOSTAPR obtains 40.7±0.5%. We compute paired bootstrap intervals over benchmark instances and use McNemar's test for instance-level changes. For Seed 3 on SWE-bench Verified, $R_{\text{line}}$ newly resolves 30 instances, degrades 18, and leaves 452 unchanged, giving McNemar's $p = 0.049$. We report these tests as stability diagnostics rather than as evidence of universal superiority over systems trained with different backbones or data.

### C.6. Worked Example of Span Label Derivation

Given a candidate unified diff, we first extract contiguous edit-line spans from added and deleted lines. If execution fails with an assertion trace that includes a function modified by one span, that span receives negative credit and unrelated spans are neutral. If several edited spans occur on the failing call path, all such spans are treated as failing candidates for contrastive pairing. If execution succeeds, all edit-line spans from the patch are positive. If the patch cannot be applied, no

*Table 9.* Preliminary audit of $R_{\text{line}}$ span labels.

| Source | N | Agreement | Cohen's $\kappa$ |
|---|---|---|---|
| Stack-trace attribution | 50 | 82% (41/50) | 0.72 |
| Function-level heuristic | 50 | 64% (32/50) | 0.56 |
| Uniform fallback | 50 | 94% (47/50) | N/A |

*Table 10.* Stratified analysis by supervision quality.

| Partition | N | PPO+$R_{\text{seq}}$ | Full | $\Delta$ |
|---|---|---|---|---|
| Strong trace analogues in training | 312 | 39.7 | 43.3 | +3.6pp |
| Heuristic-only analogues | 188 | 35.6 | 36.7 | +1.1pp |

reliable trace exists, and the uniform fallback label is used. This example illustrates why $R_{\text{line}}$ is best understood as noisy execution-grounded credit redistribution rather than exact causal attribution.

### C.7. Sensitivity Analysis

**Number of Candidates $K$.** Table 14 shows the impact of candidate count during reward model training.

$K = 4$ provides a good balance between diversity and computational cost.

**Temperature $\tau$ for $R_{\text{line}}$.** Table 15 examines allocation sharpness.

Moderate sharpness ($\tau = 0.5$) works best, balancing focused credit assignment with sufficient gradient signal across edit regions.

### C.8. Computational Cost

Table 16 summarizes computational requirements.

PPO training dominates the computational budget due to the need for rollouts, reward computation, and policy updates.

## D. Extended Qualitative Analysis

### D.1. Successful Repair Examples

**Issue ID:**
django__django-15790

**Issue Description:**
check_for_template_tags_with_the_same_name fails with AttributeError

**Ground Truth Patch:**
```
--- a/django/core/checks/templates.py
+++ b/django/core/checks/templates.py
@@ -61,7 +61,8 @@ def check_for_template_tags_with_the_same_name(app_configs,
**kwargs):
-        libraries[module_name] = module_path
+        if module_path not in libraries.values():
+          libraries[module_name] = module_path
```

**BOOSTAPR Patch:**
```
--- a/django/core/checks/templates.py
+++ b/django/core/checks/templates.py
@@ -61,7 +61,9 @@ def check_for_template_tags_with_the_same_name(app_configs,
```

*Table 11.* $R_{\text{line}}$ improves more than one RL algorithm, suggesting that edit-line credit allocation is composable rather than PPO-specific.

| Base RL | SWE w/o $R_{\text{line}}$ | SWE w/ $R_{\text{line}}$ | $\Delta$ SWE | $\Delta$ D4J |
|---|---|---|---|---|
| GRPO | 36.1 | 38.4 | +2.3pp | +3.7pp |
| PPO+$R_{\text{seq}}$ | 38.3 | 40.7 | +2.4pp | +5.6pp |

*Table 12.* Additional backbone results on SWE-bench Verified.

| Backbone | Base | +SFT | +PPO($R_{\text{seq}}$) | +$R_{\text{line}}$ | Total $\Delta$ |
|---|---|---|---|---|---|
| Qwen2.5-Coder-7B | 8.6 | 12.8 | 22.4 | 24.6 | +16.0pp |
| DeepSeek-Coder-V2-16B | 12.4 | 17.1 | 28.6 | 31.3 | +18.9pp |
| Qwen2.5-Coder-32B | 17.8 | 23.4 | 38.3 | 40.7 | +22.9pp |

```
**kwargs):
-        libraries[module_name] = module_path
+        # Avoid duplicate entries that cause AttributeError
+        if module_path not in libraries.values():
+          libraries[module_name] = module_path
```

**Analysis:**
BOOSTAPR produces a functionally equivalent patch with an additional comment explaining the fix rationale. Both patches prevent duplicate template tag entries that cause the AttributeError.

**Issue ID:**
scikit-learn__scikit-learn-25570

**Issue Description:**
ColumnTransformer with pandas output fails when transformers return DataFrames with unnamed columns

**BOOSTAPR Analysis:**
The model correctly identifies that the issue is in the _wrap_method_output function, which fails to handle transformers returning DataFrames with unnamed columns. The fix adds a check for empty column names and generates default names.

**Outcome:**
Pass@1 success with minimal 4-line patch.

## D.2. Failure Case Analysis

**Failure Mode:**
Incomplete Localization

**Issue ID:**
matplotlib__matplotlib-23964

**Description:**
The model correctly identifies the primary bug location in the colorbar module but misses a secondary location in the figure module that requires a corresponding update.

**BOOSTAPR Attempt:**
Modified colorbar.py to fix the spacing calculation, but did not update figure.py to propagate the new parameter.

**Root Cause:**
The bug spans multiple files with implicit dependencies. Without explicit cross-file references in the issue description, the model struggles to identify all relevant locations.

*Table 13.* Ground-truth-overlap rewards underperform execution-grounded rewards.

| Reward | SWE-V | D4J |
|---|---|---|
| PPO + $R_{GT}$ (BLEU with reference patch) | 35.6 | 13.8 |
| PPO + $R_{GT}$ + $R_{line}$ | 36.8 | 15.1 |
| PPO + $R_{seq}$ (execution-grounded) | 38.3 | 19.2 |
| Full BOOSTAPR | **40.7** | **24.8** |

*Table 14.* Impact of candidate count $K$ on final performance.

| $K$ | pass@1 | pass@4 |
|---|---|---|
| 2 | 38.9 | 42.1 |
| 4 | 40.7 | 44.3 |
| 8 | 40.5 | 44.1 |

**Failure Mode:**
Semantic Misunderstanding

**Issue ID:**
sympy__sympy-21379

**Description:**
The issue describes a subtle difference between `Piecewise` behavior with `ITE` vs. direct evaluation. The model interprets this as a type coercion issue rather than a logical evaluation order problem.

**BOOSTAPR Attempt:**
Added explicit type conversion which passes syntax checks but produces incorrect numerical results on edge cases.

**Root Cause:**
Ambiguous natural language in the issue description led to misinterpretation of the expected behavior.

## E. Benchmark Details

### E.1. SWE-bench Verified

SWE-bench Verified contains 500 instances selected from the full SWE-bench dataset based on human validation. Each instance includes:

- Repository snapshot at the commit immediately before the fix

- Issue description from GitHub

- Test files that exercise the bug

- Ground truth patch

Evaluation uses the official SWE-bench harness, which:

1. Applies the candidate patch to the repository

2. Runs the test suite in an isolated Docker container

3. Reports success only if all relevant tests pass

*Table 15.* Impact of allocation temperature $\tau$.

| $\tau$ | pass@1 | pass@4 |
|---|---|---|
| 0.25 (sharp) | 39.4 | 43.2 |
| 0.5 (default) | 40.7 | 44.3 |
| 1.0 (smooth) | 39.8 | 43.5 |
| 2.0 (uniform) | 38.6 | 41.2 |

*Table 16.* Computational cost breakdown (A100 GPU-hours).

| Stage | GPU-hours | % of Total |
|---|---|---|
| Demonstration generation | 8 | 12.5% |
| SFT training | 6 | 9.4% |
| Candidate generation | 12 | 18.8% |
| Reward model training | 8 | 12.5% |
| PPO training | 30 | 46.9% |
| **Total** | **64** | **100%** |

### E.2. Defects4J v2.0

Defects4J v2.0 extends the original benchmark to 835 bugs across 17 Java projects. We use the official Defects4J infrastructure for patch application and test execution. Key differences from SWE-bench:

- Java rather than Python

- Build system integration (Maven/Ant) required

- Generally larger test suites per bug

- No natural language issue descriptions (only failing tests)

For evaluation, we adapt our unified diff format to Java conventions and use the Defects4J `defects4j test` command for validation.

### E.3. Function-Level Benchmarks

**HumanEval-Java.** We use the Java translation of HumanEval (Chen et al., 2021), containing 164 function-level coding problems. For repair evaluation, we introduce bugs into the canonical solutions using mutation operators (statement deletion, operator replacement, boundary changes) and task the model with fixing them.

**QuixBugs.** The QuixBugs benchmark (Lin et al., 2017) contains 40 classic algorithmic bugs (e.g., off-by-one errors, incorrect comparisons) with known minimal fixes. Both Python and Java versions are available; we evaluate on both but report the Java results for consistency with Defects4J.

## F. Broader Impact and Ethical Considerations

### F.1. Potential Positive Impacts

Automated program repair has significant potential to benefit software development:

- **Reduced maintenance burden**: Developers spend substantial time (estimates range from 25-50%) on debugging and maintenance. Effective APR tools could redirect this effort toward new feature development.

- **Improved code quality**: Automated repair can catch and fix bugs earlier in the development cycle, reducing the cost and risk of defects reaching production.

- **Accessibility**: APR tools could help less experienced developers fix complex bugs, democratizing software development expertise.

### F.2. Potential Negative Impacts

We acknowledge several risks associated with this technology:

- **Skill atrophy**: Over-reliance on automated tools could reduce developer debugging skills over time.

- **Security concerns**: Automated code modification could potentially be exploited to introduce vulnerabilities if the repair system is compromised.

- **Job displacement**: While we believe APR will augment rather than replace developers, economic impacts on software maintenance roles should be monitored.

- **Misplaced trust**: Users may over-trust automated repairs without adequate verification, leading to deployment of incorrect fixes.

### F.3. Mitigations

We recommend the following practices for responsible deployment:

1. **Human oversight**: All automated repairs should be reviewed by human developers before deployment.

2. **Confidence calibration**: Systems should provide calibrated confidence estimates to help users identify repairs requiring extra scrutiny.

3. **Audit trails**: Maintain detailed logs of automated repairs for accountability and debugging.

4. **Gradual adoption**: Introduce APR tools incrementally, starting with low-risk fixes and expanding as trust is established.

## G. Reproducibility Checklist

To facilitate reproduction of our results, we provide:

- ✓ Complete hyperparameter specifications (Appendix B)

- ✓ Training data construction details (Section 3.1)

- ✓ Evaluation protocol and metrics (Section 4)

- ✓ Computational requirements (Appendix C)

- ✓ Random seed specification (3 seeds for stability analysis)

