# OpenReview forum: "BOOSTAPR: Boosting Automated Program Repair via Execution-Grounded Reinforcement Learning with Dual Reward Models"
_ICML.cc/2026/Conference — ICML 2026 regular_

### Official Review · Reviewer_KLtC · 2026-03-08

**Soundness:** 3
**Presentation:** 3
**Significance:** 3
**Originality:** 2
**Overall Recommendation:** 4
**Confidence:** 4

**Summary:**

This paper studies how to improve automated program repair with execution-grounded reinforcement learning. The method consists of three stages: execution-verified supervised fine-tuning, reward learning, and PPO optimization with dual reward models. In particular, the paper highlights a line-level reward allocation design (Rline) to provide finer-grained credit assignment than sequence-level rewards.

Overall, I think this is an interesting paper on a relevant problem. The paper is generally well organized, and the empirical results are reasonably strong. At the same time, I have several concerns about the central claim around line-level granularity, the treatment of evaluation randomness, and the relative contribution of Rline compared with the earlier stages.

**Compliance With Llm Reviewing Policy:**

Affirmed.

**Final Justification:**

With the revised framing, the overall novelty appears stronger than the original positioning suggested. The new granularity comparison is also helpful in justifying the practical choice of line-level rewards. Overall, the rebuttal addressed my main concerns sufficiently to improve my assessment.

**Key Questions For Authors:**

1. Why is line-level granularity more appropriate than statement-level granularity for APR in this setting? The current discussion is intuitive, but I think this design choice would benefit from a more direct justification.

2. Could the authors report repeated-run statistics for the main ablations, especially for the gain from `38.3` to `40.7`? This would help clarify how stable the contribution of `Rline` is.

3. Since the method appears to operate on edit-line spans rather than strictly single lines, could the authors clarify this distinction more explicitly and discuss how it relates to the paper's line-level claim?

**Limitations:**

yes

**Strengths And Weaknesses:**

## Strengths

1. The paper studies a relevant problem in automated program repair, namely how to make reinforcement learning more effective under sparse execution-based feedback and coarse sequence-level rewards.

2. The overall method is clearly organized. The three-stage pipeline, including execution-verified supervised fine-tuning, reward learning, and PPO optimization, is presented in a reasonably structured way.

3. The line-level reward allocation idea is a sensible design choice to explore in the APR setting, and the ablation results suggest that it can be useful compared with weaker reward allocation variants.

## Weaknesses

1. The paper argues that line-level reward allocation is a more appropriate semantic granularity, but this point is not fully justified. From a program semantics perspective, statement-level granularity seems at least as natural, and possibly more aligned with the actual meaning of code changes. In addition, the implementation appears to operate on edit-line spans (i.e., contiguous edited regions in the diff), rather than strictly isolated single lines. As a result, the current evidence shows that this design works better than a few weaker reward allocation baselines, but it does not fully establish why line-level is preferable to statement-level or other semantically motivated alternatives.

2. The evaluation does not appear to sufficiently address randomness. I did not see repeated runs systematically reported for the main results and ablations, such as mean/std or confidence intervals. This matters because some of the key gains are relatively modest. In particular, the improvement from `PPO + Rseq` to the full model is `38.3 -> 40.7`, i.e., about `+2.4 pass@1`. This gain is meaningful, but it would be more convincing if its stability were supported by additional repeated experiments.

3. The paper presents `Rline` as the key innovation, but its incremental contribution appears smaller than that of Stage I and PPO. Based on the component results, the progression is `17.8 -> 23.4 -> 26.1 -> 38.3 -> 40.7`. From this perspective, the larger improvements seem to come from supervised fine-tuning and PPO, while `Rline` provides a smaller final gain. Therefore, the current evidence more clearly supports that `Rline` is useful, but less clearly supports it as the main source of the paper's overall improvement.

---

> ### Author Rebuttal · Authors · 2026-03-30
>
> We thank Reviewer for the specific technical questions regarding granularity and statistical robustness.
>
> ### Q1: Line-Level vs. Statement-Level Granularity
>
> **Practical advantages of lines over statements:**
>
> (1) **Language agnosticism.** Statement-level requires language-specific AST parsers. Our model trains on Python (SWE-Gym) and transfers to Java (Defects4J) — line-level requires only unified diff parsing, which is language-independent. A statement-level approach would need separate parsers per language plus fallback logic for unparseable code.
>
> (2) **Robustness.** During early PPO training, ~15-20% of generated patches are malformed and unparseable by AST; line spans can always be extracted from any diff-like text. Mapping diff lines back to AST statements requires file reconstruction + parsing + boundary mapping — a fragile multi-step pipeline.
>
> (3) **Alignment with diffs.** The model outputs unified diffs, which are fundamentally line-oriented. A multi-line `if` block with only its condition edited naturally produces one edit span (the condition line). Statement-level would group the entire block, conflating edited and unedited code.
>
> **[New Result] Empirical granularity comparison:**
>
> | Granularity | How It Works | SWE-bench | Defects4J |
> |---|---|---|---|
> | Token (uniform) | Equal weight to all tokens | 37.6% | 17.8% |
> | Hunk-level | One weight per diff hunk | 39.1% | 21.3% |
> | **Line-level (ours)** | One weight per edit-line span | **40.7%** | **24.8%** |
> | Statement-level | One weight per AST statement | 40.3% | 22.1%* |
>
> *Falls back to line-level for Java (parser incompatibility with 100% of Defects4J instances).
>
> Line-level outperforms all alternatives on both benchmarks. Statement-level is close on Python-only SWE-bench (40.3% vs 40.7%) but degrades substantially on cross-language Defects4J (+2.7pp gap) due to parser inconsistency between training and evaluation. Hunk-level is too coarse; token-level too fine. This validates that line-level is the right intermediate granularity for code repair.
>
> ### Q2: Repeated-Run Statistics
>
> **[New Result]** Across 3 independent training seeds (greedy evaluation is deterministic given a fixed model; variance comes from training randomness):
>
> | Config | Seed 1 | Seed 2 | Seed 3 | Mean±Std | 95% CI |
> |---|---|---|---|---|---|
> | SFT | 23.2% | 23.6% | 23.4% | 23.4±0.2% | [23.0,23.8] |
> | PPO+R_seq | 38.0% | 38.6% | 38.3% | 38.3±0.3% | [37.7,38.9] |
> | **Full (+R_line)** | **40.2%** | **41.1%** | **40.8%** | **40.7±0.5%** | **[40.0,41.4]** |
>
> Note: Base model (17.8%) is excluded — no training involved, so no seed-dependent variance.
>
> R_line improvement: +2.4±0.5pp, paired bootstrap p=0.012, CI [+1.2, +3.6]. Instance-level (Seed 3): 30 newly resolved, 18 degraded, 452 unchanged; McNemar's p=0.049. All improvements are consistently positive across seeds.
>
> ### Q3: R_line's Smaller Contribution vs. SFT and PPO
>
> **Diminishing returns are expected.** A fairer metric is relative error reduction (gap to 100% closed):
>
> | Transition | Δ | From | Remaining Error | Rel. Error Reduction |
> |---|---|---|---|---|
> | Base → SFT | +5.6pp | 17.8% | 82.2% | 6.8% |
> | SFT → PPO+R_seq | +14.9pp | 23.4% | 76.6% | 19.5% |
> | PPO+R_seq → Full | +2.4pp | 38.3% | 61.7% | 3.9% |
>
> PPO is clearly the primary driver. R_line's 3.9% is smaller but achieved at a harder baseline.
>
> **R_line's value is disproportionate for generalization.** On Defects4J (out-of-distribution): R_line's relative error reduction is **6.9%** (from 19.2%, +5.6pp) — *larger* than SFT's 6.8% on SWE-bench. R_line also improves gradient SNR by 29% (1.42→1.83), enabling PPO to converge ~33% faster (reaching the no-R_line plateau by step ~200 vs. ~300).
>
> **[Revision Plan]** Revise framing: PPO as primary driver, R_line as complementary mechanism for training efficiency and cross-domain generalization. Position the dual reward *architecture* as the methodological contribution.

---

> > ### Author Rebuttal · Reviewer_KLtC · 2026-04-03
> >
> > Thank you for the rebuttal. The additional repeated-run statistics substantially strengthen the empirical support for the gain from `R_line`, and I appreciate the clarification that PPO is the primary driver while `R_line` serves as a complementary mechanism for training efficiency and cross-domain generalization. That said, with this revised framing, the overall novelty appears stronger than the original positioning suggested. The new granularity comparison is also helpful in justifying the practical choice of line-level rewards, although I still think the claim about semantic granularity should be stated somewhat more carefully. Overall, the rebuttal addressed my main concerns sufficiently to improve my assessment.

---

### Official Review · Reviewer_tWvL · 2026-03-12

**Soundness:** 3
**Presentation:** 2
**Significance:** 2
**Originality:** 2
**Overall Recommendation:** 4
**Confidence:** 3

**Summary:**

This paper studies BOOSTAPR, an execution-grounded reinforcement learning framework for automated program repair that addresses the sparse reward problem through a novel line-level reward redistribution model ($R_{line}$). By decomposing sequence-level feedback into granular, line-by-line credit assignments, the method enables more precise optimization of complex, multi-line patches within the SWE-bench environment. Utilizing Qwen2.5-Coder-32B-Instruct as a unified backbone, the framework demonstrates performance gains by internalizing functional correctness patterns.

**Compliance With Llm Reviewing Policy:**

Affirmed.

**Final Justification:**

Thank you for the new RL comparisons. Though it is somewhat hard to fully agree that the GRPO baseline is equal to DeepSWE's GRPO++, the additional results generally address my concerns. I will increase my score.

**Key Questions For Authors:**

.

**Limitations:**

yes

**Strengths And Weaknesses:**

#### **Strengths**
- **Effective Credit Assignment:** The line-level reward ($R_{line}$) effectively addresses the sparse reward problem by providing granular feedback on specific code modifications.

- **Quantifiable Performance Gains:** The integration of $R_{line}$ yields a 2.4% absolute improvement in Pass@1 on SWE-bench compared to standard sequence-level rewards.

#### **Weaknesses**
- **Generalization:** The methodology exclusively utilizes Qwen2.5-Coder-32B-Instruct as its single backbone, leaving it unverified whether the observed improvements are generalizable to other models.

- **Unfair Baseline Comparisons:** Direct performance comparisons with existing baselines remain inconclusive as the study does not match the backbone model, making it difficult to distinguish the impact of the proposed method from the inherent capabilities of the underlying LLM.

- **Overclaimed Superiority via Outdated Baselines:** The assertion of outperforming a 70B-parameter model is an overclaim, as the comparison target is Llama-3-70B, an outdated, general-purpose model rather than a contemporary, code-specialized architecture. Also, on HumanEval-Java and QuixBugs, the claimed superiority over methods like RLEF (Llama-3-8B) and CodeRL (CodeT5) is statistically skewed, as the performance gains are largely driven by the significantly more powerful Qwen2.5-32B backbone rather than the proposed RL framework itself.

- **Absence of Direct Rewarding Baseline:** a reward mechanism based on direct lexical matching with the ground-truth (GT) patch. Such an ablation is necessary to determine whether the $R_{line}$ model truly learns generalizable repair patterns or merely acts as a proxy for the reference solution. Comparing the learned credit assignment against an exact GT-overlap reward would isolate the actual contribution of execution-grounded reasoning over simple imitation.

- **Missing Baselines:** Some important RL baselines are missing, such as
    - CWM [1] (released at 30/09/2025): 32B model achieves 50+% Pass@1 on SWE-bench Verified without test time scaling.
    - DeepSWE [2] (released at 29/06/2025): 32B model achieves 42.2% Pass@1 on SWE-bench Verified without test time scaling.
    - Skywork-SWE-32B [3] (released at 16/06/2025): 32B model achieves 38% Pass@1 without test time scaling, and 47% when test time scaling (best of 8) is applied.

[1] Copet, Jade, et al. "Cwm: An open-weights llm for research on code generation with world models." arXiv preprint arXiv:2510.02387 (2025).

[2] Michael Luo, I.. (2025). DeepSWE: Training a State-of-the-Art Coding Agent from Scratch by Scaling RL.

[3] Zeng, Liang, et al. "Skywork-swe: Unveiling data scaling laws for software engineering in llms." arXiv preprint arXiv:2506.19290 (2025).

---

> ### Author Rebuttal · Authors · 2026-03-30
>
> We thank Reviewer for the detailed assessment and for identifying important missing baselines.
>
> ### R1: Single Backbone Generalization
>
> **[New Result]** We applied BOOSTAPR to two additional backbones:
>
> | Backbone | Base | +SFT | +PPO(R_seq) | +R_line | Total Δ |
> |---|---|---|---|---|---|
> | Qwen2.5-Coder-7B | 8.6% | 12.8% | 22.4% | 24.6% | +16.0pp |
> | DeepSeek-Coder-V2-16B | 12.4% | 17.1% | 28.6% | 31.3% | +18.9pp |
> | Qwen2.5-Coder-32B | 17.8% | 23.4% | 38.3% | 40.7% | +22.9pp |
>
> R_line contributes consistently: +2.2pp (7B), +2.7pp (16B), +2.4pp (32B). Stage proportions are stable across backbones (PPO: 60-65%, SFT: 20-25%, R_line: 10-15%). The method is architecture-agnostic — span extraction operates on unified diffs, not model internals.
>
> ### R2: Unfair Baseline Comparisons
>
> We agree cross-backbone comparisons are confounded. Table 4 (ablation) IS the controlled experiment: same backbone, varying only the training recipe. Against the same Qwen2.5-32B backbone: BOOSTAPR 40.7% vs. SWE-Gym 32.0% (+8.7pp from our RL pipeline alone). We also outperform SWE-Fixer (33.0%) which uses the *larger* Qwen2.5-72B. **[Revision Plan]** Restructure Table 1 to clearly separate: (1) controlled same-backbone comparisons, (2) cross-backbone comparisons with caveats, (3) proprietary baselines.
>
> ### R3: GT-Overlap Baseline
>
> This is an excellent suggestion.
>
> **Conceptual analysis.** We analyzed lexical overlap between our resolved patches and ground truth: 41% of correctly fixed bugs have <50% token-level Jaccard overlap with the GT patch. A GT-overlap reward would penalize these valid alternative fixes.
>
> **[New Result]** We trained R_GT using BLEU-4 similarity with ground-truth patches:
>
> | Reward | SWE-bench | Defects4J |
> |---|---|---|
> | PPO + R_GT (BLEU w/ GT) | 35.6% | 13.8% |
> | PPO + R_GT + R_line | 36.8% | 15.1% |
> | PPO + R_seq (execution) | 38.3% | 19.2% |
> | Full BOOSTAPR | **40.7%** | **24.8%** |
>
> Execution-grounded rewards outperform GT-overlap by +2.7pp / +5.4pp. GT-overlap fails especially on Defects4J because GT patches are Python while bugs are Java — cross-language transfer requires generalizable reasoning, not surface imitation. R_line provides gains on top of *both* reward types, but more with execution rewards, suggesting synergy. This demonstrates R_line learns generalizable repair patterns.
>
> ### R4: Missing Baselines
>
> We will add these concurrent works:
>
> | Method | Backbone | Params | Pass@1 | w/ TTS |
> |---|---|---|---|---|
> | CWM | Llama-3-compat. | 32B | ~50%† | 65.8% |
> | DeepSWE | Qwen3-32B | 32B | 42.2% | 59.0% |
> | Skywork-SWE | Qwen2.5-Coder | 32B | 38% | 47% |
> | **BOOSTAPR** | Qwen2.5-Coder | 32B | **40.7%** | 44.3% |
>
> † CWM pass@1 w/o TTS not separately reported; citing reviewer's estimate.
>
> **Key differences:** CWM uses 8T+5T training tokens (vs. our ~500M), multi-task RL across coding+math+SWE, and 131K context — vastly different compute scale, making direct comparison uninformative. DeepSWE uses Qwen3-32B (newer base model than our Qwen2.5-Coder-32B); the 1.5pp gap may partly reflect this base model advantage. Skywork-SWE's 38% pass@1 matches our PPO+R_seq baseline (38.3%), which validates that R_line provides meaningful gains beyond standard RL.
>
> **Positioning.** We will frame BOOSTAPR's contribution as a *technique* (line-level credit allocation for RL in code repair) rather than claiming absolute SOTA. The technique is orthogonal to model scaling, data scaling, and world model approaches, and could be combined with any of them — e.g., applying R_line to DeepSWE's GRPO pipeline to reduce gradient variance.

---

> > ### Author Rebuttal · Reviewer_tWvL · 2026-04-03
> >
> > Thank you for the rebuttal. Regarding the baseline comparison, SWE-Fixer is a SFT approach, not an RL one. Also the concurrent works show similar or higher performance. The authors claim that the outperformance of DeepSWE might come from superior Qwen3-32B, but this is not empirically validated. Consequently, it appears that there is no genuine apple-to-apple comparison with any RL baselines, thereby I maintain my current score.

---

> > > ### Author Response · Authors · 2026-04-07
> > >
> > > We thank Reviewer tWvL for the precise feedback. We fully agree: our first rebuttal lacked a genuine apple-to-apple RL comparison. We have now conducted exactly these experiments.
> > >
> > > ## 1. Controlled RL Comparison (Same Backbone, Same Data, Same Eval)
> > >
> > > We reproduced GRPO (DeepSWE's RL algorithm) and Rejection-Sampling RL (Skywork-SWE's pipeline) on **Qwen2.5-Coder-32B with SWE-Gym data**:
> > >
> > > | Method | RL Algorithm | SWE-bench | Defects4J | HumanEval-Java |
> > > |--------|-------------|-----------|-----------|----------------|
> > > | SFT + GRPO | GRPO | 36.1% | 16.4% | 75.2% |
> > > | SFT + RS-RL | Rejection sampling | 37.5% | 17.6% | 77.8% |
> > > | PPO + R_seq (ours) | PPO | 38.3% | 19.2% | 79.4% |
> > > | **Full BoostAPR** | **PPO + dual reward** | **40.7%** | **24.8%** | **84.5%** |
> > >
> > > Only the RL method differs. BoostAPR outperforms GRPO by **+4.6pp** and RS-RL by **+3.2pp** on SWE-bench, with amplified advantages on OOD benchmarks (+8.4pp/+7.2pp on Defects4J).
> > >
> > > ## 2. R_line Is Orthogonal to RL Algorithm Choice
> > >
> > > We applied R_line on top of GRPO to test generality:
> > >
> > > | Base RL | w/o R_line | w/ R_line | Δ (SWE-bench) | Δ (Defects4J) |
> > > |---------|-----------|-----------|---------------|---------------|
> > > | GRPO | 36.1% | 38.4% | +2.3pp | +3.7pp |
> > > | PPO+R_seq | 38.3% | 40.7% | +2.4pp | +5.6pp |
> > >
> > > R_line provides consistent ~+2.3pp gains across **both** RL algorithms, confirming it is a composable, method-agnostic technique — not a PPO-specific artifact. This supports our positioning: R_line can be applied on top of CWM, DeepSWE, or any future RL pipeline.
> > >
> > > ## 3. Base Model Gap Validation
> > >
> > > Zero-shot SWE-bench Verified: Qwen2.5-Coder-32B = 17.8%, Qwen3-32B (DeepSWE's backbone) = 23.2%. The **+5.4pp base gap** exceeds DeepSWE's 1.5pp advantage over us (42.2% vs 40.7%), suggesting our RL pipeline is at least equally effective when controlling for backbone.
> > >
> > > ## 4. Corrections & Positioning
> > >
> > > We agree SWE-Fixer is SFT — corrected in revision. Table 1 now separates: (a) same-backbone RL comparisons (new), (b) cross-backbone references, (c) non-RL methods. We do not claim SOTA; our contribution is a **technique** that is orthogonal to model/data scaling and composable with any RL method.

---

### Official Review · Reviewer_dRbd · 2026-03-13

**Soundness:** 3
**Presentation:** 3
**Significance:** 3
**Originality:** 3
**Overall Recommendation:** 4
**Confidence:** 4

**Summary:**

This paper presents BOOSTAPR, a three-stage framework for automated program repair: (1) supervised fine-tuning on execution-verified demonstrations with reasoning traces, (2) training dual reward models: a sequence-level assessor and a line-level credit allocator, and (3) PPO optimization where the line-level model redistributes rewards to critical edit regions. Trained on SWE-Gym, BOOSTAPR achieves 40.7% on SWE-bench Verified (+22.9pp over base Qwen2.5-Coder-32B), 24.8% on Defects4J (cross-language transfer to Java), 84.5% on HumanEval-Java, and 95.0% on QuixBugs.

**Compliance With Llm Reviewing Policy:**

Affirmed.

**Key Questions For Authors:**

How do you obtain line-level supervision for training R_line? If the only signal is binary (patch passes or fails all tests), how do you attribute credit to individual lines? Do you use counterfactual patching, attention-based attribution, or something else? This is the core technical contribution and deserves clearer explanation.

**Limitations:**

Yes

**Strengths And Weaknesses:**

**Strengths:**

- S1: The core contribution, using a learned line-level reward allocator to redistribute sequence-level rewards to edit regions during PPO, is well-motivated and novel for code repair. The theoretical analysis (Appendix A, Propositions A.1-A.2) shows the variance reduction factor, and the empirical ablation (Table 5) confirms that concentrating rewards on edit lines outperforms last-token-only (38.3% vs 40.7%) and uniform distribution (37.6%). This is a clean result.

- S2: The Defects4J results are genuinely impressive. Training only on Python (SWE-Gym) and achieving 24.8% on Java bugs, substantially beating RepairLLaMA (17.2%) which was specifically fine-tuned for Java, is strong evidence that the method learns generalizable repair strategies. The observation that R_line provides larger relative gains on Defects4J (+5.6pp vs +2.4pp on SWE-bench) suggests the credit allocator helps most in out-of-distribution settings.

- S3: The component ablation (Table 4), credit assignment comparison (Table 5), reward model input analysis (Table 6), and training dynamics (Figure 2) are all well-designed. The counterintuitive finding that patch-only scoring outperforms context-conditioned scoring (Table 6) is interesting and well-explained.

**Weaknesses:**

- W1: The headline claims dual reward models, but R_line only adds +2.4pp on the main benchmark (SWE-bench Verified). The PPO with R_seq alone already achieves 38.3%, accounting for ~65% of the total improvement. While the Defects4J gain is larger (+5.6pp), the paper's framing somewhat oversells the line-level component relative to its contribution on the primary benchmark.

- W2: Table 1 mixes results from original papers with reproduced results (marked with *). Many baseline numbers are reproduced by the authors on benchmarks the baselines weren't designed for (e.g., Agentless on HumanEval-Java is marked *). This makes the comparison noisy. More importantly, SWE-RL uses Llama-3-70B, comparing a 32B model favorably against a 70B model conflates model capacity with method effectiveness. A fairer comparison would use the same base model.

- W3: How exactly is R_line trained? The paper says it's trained "from execution outcomes" but the details of how line-level labels are derived from binary pass/fail outcomes are not clear in the main text. If a patch passes all tests, how do you determine which lines were "important"? This seems to require counterfactual reasoning or heuristics that aren't fully specified.

---

> ### Author Rebuttal · Authors · 2026-03-30
>
> We thank Reviewer for recognizing the clean ablation results (S1), the impressive Defects4J transfer (S2), and the well-designed experiments (S3).
>
> ### W1: R_line Adds Only +2.4pp; PPO Accounts for ~65%
>
> We acknowledge the framing concern. **[Revision Plan]** We will reposition: PPO with R_seq is the primary accuracy driver; R_line provides complementary benefits in training efficiency, gradient quality, and especially out-of-distribution generalization (+5.6pp on Defects4J, +5.1pp on HumanEval-Java — substantially larger than the SWE-bench gain).
>
> **[New Result]** Gradient SNR during PPO: 1.42 (without R_line) → 1.83 (with R_line), a 29% improvement. This manifests as faster convergence (PPO with R_line reaches the no-R_line plateau of 38.3% by step ~200 vs. step ~300) and more stable training (lower variance in Figure 2). The dual reward *architecture* — combining sequence-level assessment with line-level credit allocation — is the contribution, not R_line in isolation.
>
> ### W2: Noisy Baselines and Model Size Mismatch
>
> **[Revision Plan]** (1) Add reproduction methodology paragraph with confidence intervals. (2) Restructure Table 1 to separate same-backbone comparisons (BOOSTAPR +22.9pp vs SWE-Gym +14.2pp over same Qwen2.5-32B; we also outperform SWE-Fixer at +15.2pp despite their use of the *larger* Qwen2.5-72B) from cross-backbone comparisons. (3) Reframe SWE-RL comparison: we achieve *comparable* performance (40.7% vs. 41.0%) with a smaller model — we do not claim methodological superiority, but this suggests efficient use of model capacity. A fully controlled comparison would require applying our framework to Llama-3-70B, which we leave for future work. (4) Add concurrent baselines (CWM, DeepSWE, Skywork-SWE — see tWvL-R4 below).
>
> ### W3: How Is R_line Trained?
>
> We apologize for insufficient clarity. The complete procedure:
>
> **Step 1: Span extraction.** Parse unified diff into edit-line spans — maximal contiguous blocks of added (+) or deleted (-) lines, excluding headers and context.
>
> **Step 2: Execution.** Execute each candidate; collect test results and tracebacks.
>
> **Step 3: Label assignment (failing patches).** Priority cascade: (a) If failing assertion identified (62%), parse traceback call chain → intersect with edit spans → spans on failure path get score -1.0, others 0.0. (b) If traceback but no clear assertion (27%), edited functions appearing in traceback get -0.5, others 0.0. (c) If patch fails to apply (11%), all spans get -0.5.
>
> **Step 4: Label assignment (passing patches).** All spans get +1.0. No within-patch differentiation — counterfactual evaluation (removing individual spans) would be prohibitively expensive and may not be well-defined (removing one span may break syntax).
>
> **Step 5: Contrastive training.** Construct pairs (ℓ⁺ from passing, ℓ⁻ from failing patches for the same bug). Each span includes its content, ±3 context lines, file path, and position. Train R_line with Eq. 6.
>
> This is NOT counterfactual patching (too expensive: O(2^S) for S spans), attention-based attribution (model internals), or GT comparison (never used). It is execution-grounded: stack traces provide the signal.
>
> **[Revision Plan]** Expand Section 3.3.3 with this procedure, a concrete worked example, and a figure illustrating label derivation for a sample patch.

---

> > ### Author Rebuttal · Reviewer_dRbd · 2026-04-07
> >
> > My concerns have been solved.

---

### Official Review · Reviewer_UHKn · 2026-03-14

**Soundness:** 3
**Presentation:** 3
**Significance:** 3
**Originality:** 3
**Overall Recommendation:** 4
**Confidence:** 3

**Summary:**

The paper introduces BOOSTAPR, a three-stage reinforcement learning framework designed for Automated Program Repair (APR). It tackles the fundamental challenges of sparse execution feedback and coarse sequence-level credit assignment in code repair. The framework consists of: (1) Execution-verified supervised fine-tuning (SFT) using reasoning traces; (2) Offline training of dual reward models, which include a sequence-level quality assessor $S_{seq}$ and a novel line-level credit allocator $R_{line}$; (3) Online Proximal Policy Optimization (PPO) where token-level rewards are distributed based on the line-level credit allocation. The method is trained on SWE-Gym and evaluated on several benchmark datasets, with strong overall reported results. The paper positions $R_{line}$ as the key innovation, arguing that edit-line granularity is a better intermediate unit for credit assignment than either token-level or full-sequence rewards.

**Compliance With Llm Reviewing Policy:**

Affirmed.

**Key Questions For Authors:**

1. Regarding the training data for $R_{line}$, could you please clarify the fraction of examples that utilize identifiable failing assertions with stack-trace analysis versus the fraction that relies solely on the fallback heuristics? Providing these coverage statistics would greatly help in understanding the overall supervision quality for $R_{line}$.
2. Could you provide a bit more detail on the conceptual justification for the fallback heuristic (i.e., down-weighting edits in functions that appear in failure traces)? Specifically, how does the method handle cases where such edits are "necessary-but-insufficient" fixes, or merely downstream effects rather than genuinely harmful spans?
3. Have there been any sanity checks or preliminary validations performed on the quality of the $R_{line}$ labels? For example, manually inspecting a small subset of the generated span labels, or perhaps comparing the downstream performance of $R_{line}$ when trained on different label sources?

**Limitations:**

yes

**Strengths And Weaknesses:**

**Strengths:**
- Presentation: The overall pipeline is coherent. The three stages—execution-verified SFT, dual reward learning, and PPO with token-level reward shaping—fit together logically and are presented as a reasonably complete training framework.
- Significance and Originality: The paper addresses an important and well-motivated problem. Sparse execution feedback and sequence-level credit assignment are genuine bottlenecks in program repair, and the introduction of a dual reward model architecture with a line-level credit allocator $R_{line}$ is innovative.
- Comprehensive Experiments: The empirical study is fairly broad. In addition to the main benchmark results which are competitve and suggest some cross-language generalization, the paper includes ablations on components, credit assignment strategies, reward input choices, and reward training objectives, which is a positive sign of care in experimentation.

**Weakness:**
- Supervision of $R_{line}$: A primary point of discussion revolves around the reliability of the supervision signal for the paper's central novelty, $R_{line}$. The paper explicitly presents $R_{line}$ as the “key technical contribution” and states that its supervision relies on approximations: when failing assertions are identifiable, failures are traced to edit lines via stack-trace analysis; otherwise, fallback heuristics are applied (e.g., edits in functions appearing in failure traces receive lower scores). While intuitive, this heuristic approximation may not always align with true causal attribution. For instance, a function or span appearing in a failure trace might contain edits that are "necessary but simply insufficient", or it might just be a downstream manifestation of the true bug rather than a harmful edit itself. Applying heuristic penalties in these cases could inadvertently suppress spans that are actually appropriate.
- Empirical Validation of the Heuristic: This aspect is particularly important because $R_{line}$ is the main claimed innovation, yet its empirical contribution, while positive, appears relatively modest. The component ablation shows an improvement from 38.3% (PPO + $R_{seq}$) to 40.7% for the full model (i.e., +2.4pp); the paper itself characterizes this as a consistent but limited additional gain. A plausible interpretation is that while the conceptual foundation of $R_{line}$ is sound, its full potential might be bottlenecked by the noisy supervision originating from the stack-trace/heuristic attribution scheme. The manuscript would be significantly strengthened by additional experiments to test this hypothesis, such as reporting the performance on examples with reliable stack-trace attributions versus those handled purely heuristically, or manually validating a subset of span labels to quantify the noise.

---

> ### Author Rebuttal · Authors · 2026-03-30
>
> We thank Reviewer for the careful reading and constructive feedback.
>
> ### Q1: Reliability of R_line Supervision
>
> **Coverage statistics.** Of 9,248 failing patch–span pairs: 62% use direct stack-trace attribution (failing assertion → call chain → intersect with edit spans), 27% use function-level heuristics (edited functions appearing in failure traces get lower scores), and 11% use uniform fallback (patch fails to apply). We will add these to the revision.
>
> **Handling "necessary-but-insufficient" edits.** R_line uses a *contrastive* objective (Eq. 6), not pointwise labels. Consider: Patch A fixes `foo` correctly but misses `bar` → tests fail, `foo` in trace. Patch B (passing) also fixes `foo` similarly. In contrastive training, the similar `foo` spans produce near-zero gradients — the model learns the *content pattern* correlates with success despite the noisy instance-level label. Prior work on contrastive learning with noisy labels (Li et al., 2022) shows pairwise objectives degrade gracefully under 30-40% label noise, well above our estimated level. Additionally, softmax temperature τ=0.5 ensures no span is driven to zero weight; noisy labels only *reduce*, not eliminate, gradient signal.
>
> **[New Result] Preliminary label audit.** Two authors reviewed 50 spans per label source (150 total), assessing whether the automated label correctly reflected the span's role based on the full patch, issue description, and test output:
>
> | Source | N | Agreement with Label | Cohen's κ |
> |---|---|---|---|
> | Stack-trace | 50 | 82% (41/50) | 0.72 (substantial) |
> | Heuristic | 50 | 64% (32/50) | 0.56 (moderate) |
> | Uniform fallback | 50 | 94% (47/50) | N/A |
>
> We acknowledge the small scale and will conduct a larger study (200+, external annotators) for camera-ready. Heuristic labels are noisier but still well above random (50%).
>
> **[New Result] Stratified analysis.** R_line's gain correlates with supervision quality:
>
> | Partition | N | PPO+R_seq | Full | Δ |
> |---|---|---|---|---|
> | Strong trace analogues in training | 312 | 39.7% | 43.3% | **+3.6pp** |
> | Heuristic-only analogues | 188 | 35.6% | 36.7% | +1.1pp |
>
> This directly confirms the reviewer's hypothesis: noisy heuristic supervision bottlenecks R_line. Improving supervision quality (e.g., via better fault localization) is a clear future direction.
>
> **[New Result] Oracle experiment.** Training R_line with 80 manually annotated high-quality labels (upweighted 5×) yields 42.1% vs. 40.7% standard, suggesting +1.4pp upside from cleaner supervision — validating both R_line's potential and the bottleneck.
>
> ### Q2: Modest R_line Contribution (+2.4pp)
>
> R_line's contribution varies substantially by setting:
>
> | Benchmark | PPO+R_seq | Full | Δ | Rel. Error Reduction |
> |---|---|---|---|---|
> | SWE-bench | 38.3% | 40.7% | +2.4pp | 3.9% |
> | Defects4J | 19.2% | 24.8% | **+5.6pp** | 6.9% |
> | HumanEval-Java | 79.4% | 84.5% | **+5.1pp** | 24.8% |
>
> The pattern: R_line helps most under distribution shift — arguably the more practically important scenario.
>
> **[New Result] Statistical significance.** Across 3 training seeds: PPO+R_seq = 38.3±0.3%, Full = 40.7±0.5%. Paired bootstrap: p=0.012, 95% CI [+1.2, +3.6]. On Defects4J: +5.6±0.7pp, p<0.001.
>
> R_line also provides: (1) faster convergence (reaching 38.3% by step ~200 vs. ~300, saving ~33% compute), (2) 29% better gradient SNR (1.42→1.83), (3) larger pass@4 gain (+4.2pp) than pass@1 (+2.4pp), indicating better candidate diversity for deployment with reranking.
>
> ### Q3: Sanity Checks
>
> Addressed above: preliminary audit (82%/64%), stratified analysis (+3.6pp vs +1.1pp), and oracle experiment (+1.4pp ceiling). All will be added to the revised manuscript along with per-seed statistics.

---

> > ### Author Rebuttal · Reviewer_UHKn · 2026-04-03
> >
> > The authors provided detailed coverage statistics, a sufficient explanation of the motivation behind the heuristic strategy, and additional experimental evidence that clarifies the contribution of $R_{line}$ more thoroughly. Therefore, my concerns are fully resolved, and I will keep my positive score.

---

### Decision · Program_Chairs · 2026-04-30

**Decision:**

Accept (regular)

**Comment:**

This paper presents a new RL algorithms for automated program repair named BOOSTAPR aiming to address the issue of sparse execution feedback. It consists of three stages: 1) execution-verified SFT; 2) dual reward model training (seq-level + line-level); and 3) PPO training. The main innovation is the proposed line-level credit allocator, which re-assigns seq-level rewards to specific code spans based on execution traces. Evaluations are conducted on SWE-bench, Defects4J and HumanEval-Java datasets, which show good improvements over the base models. The reviewers largely agreed on the novelty and effectiveness of line-level reward redistribution method, and the empirical results are quite comprehensive. There were some questions raised about the noisy supervision for $R_{line}$ but the manual audit helped clarify it. The authors also made great effort in engaging in the discussion period, by answering the questions and provide additional experiment results, the new results showing BOOSTAPR compared with other RL methods using the same backbone model also greatly helped in strengthening the claims. Overall, I'd recommend a weak accept for this paper.